# Ultra-wide bandgap semiconductor Ga$_2$O$_3$ power diodes

Jincheng Zhang[1], Pengfei Dong[1], Kui Dang [1], Yanni Zhang[1], Qinglong Yan[1], Hu Xiang[1], Jie Su[1], Zhihong Liu[1], Mengwei Si [2], Jiacheng Gao[3], Moufu Kong[3], Hong Zhou [1✉] & Yue Hao[1]

Ultra-wide bandgap semiconductor Ga$_2$O$_3$ based electronic devices are expected to perform beyond wide bandgap counterparts GaN and SiC. However, the reported power figure-of-merit hardly can exceed, which is far below the projected Ga$_2$O$_3$ material limit. Major obstacles are high breakdown voltage requires low doping material and PN junction termination, contradicting with low specific on-resistance and simultaneous achieving of n- and p-type doping, respectively. In this work, we demonstrate that Ga$_2$O$_3$ heterojunction PN diodes can overcome above challenges. By implementing the holes injection in the Ga$_2$O$_3$, bipolar transport can induce conductivity modulation and low resistance in a low doping Ga$_2$O$_3$ material. Therefore, breakdown voltage of 8.32 kV, specific on-resistance of 5.24 mΩ·cm$^2$, power figure-of-merit of 13.2 GW/cm$^2$, and turn-on voltage of 1.8 V are achieved. The power figure-of-merit value surpasses the 1-D unipolar limit of GaN and SiC. Those Ga$_2$O$_3$ power diodes demonstrate their great potential for next-generation power electronics applications.

[1] State Key Discipline Laboratory of Wide Bandgap Semiconductor Technology, School of Microelectronics, Xidian University, Xi'an 710071, China. [2] Department of Electronic Engineering, Shanghai Jiao Tong University, Shanghai 200240, China. [3] State Key Laboratory of Electronic Thin Films and Integrated Devices of China, University of Electronic Science and Technology of China, Chengdu 61005, China. ✉email: hongzhou@xidian.edu.cn

Advanced semiconductor material holds great promise of providing higher conversion efficiency as well as maintaining higher voltage for modern industrial- and consumer-scale power electronics. Ultra-wide bandgap (UWB) semiconductor with general bandgap ($E_g$) greater than 4 eV can sustain a higher critical field ($E_c$) and hence a higher blocking voltage is achievable at a smaller resistance and power electronic component dimension, which turns out to be more efficient than its narrow bandgap material Si and wide bandgap material GaN and SiC counterparts, as summarized in Table 1 of Supplementary Information. The general concept lies behind is that the high electric-field ($E$) and high temperature driven of the electron excitation from valance band to conduction band is inherently suppressed by the UWB. Therefore, power electronics based on UWB materials are spontaneously endowed with high breakdown voltage (BV) at a lower material thickness and resistance. Combined with good mobility ($\mu$), a crucial power device parameter Baliga's figure-of-merit (B-FOM~$\mu \times E_c^3$) of UWB semiconductors could be several or tens of folds of those wide bandgap materials GaN and SiC as well as more than thousands of times of narrow bandgap material Si[1,2]. However, it should be noted that the major tyranny of the UWB forbids achieving effective both n- and p-type doping simultaneously. Among those intriguing UWB semiconductor materials, the emerging $Ga_2O_3$ is now regarded as one of the most promising materials for next-generation high-power and high-efficiency electronics, due to its cost-effective melt-grown large-scale and low defect density substrate as well as the controllable n-type doping[3].

$Ga_2O_3$ with $E_g = 4.6$–$4.8$ eV, high $E_C = 8$ MV/cm and decent intrinsic $\mu = 250$ cm$^2$/Vs has yielded a B-FOM to be around 3000, which is four times GaN and ten times SiC. Being the mainstream of the UWB semiconductor, $Ga_2O_3$-based power electronics are expected to bring higher blocking voltage at a lower specific on-resistance ($R_{on,sp}$) for power switching applications. Tremendous efforts have been dedicated to explore the material property and push the device limit, and hence significant progresses are acquired during the past 5 years. Despite those intriguing achievements, it should be noted that those performances especially the representative device parameter power figure-of-merit (P-FOM = BV$^2$/$R_{on,sp}$) are much inferior to the projected material limit, or even cannot be comparable with the 1-D unipolar limit of the GaN and SiC[4,5]. Like other UWB semiconductors with the difficulties of achieving both highly conductive p- and n-type materials at the meantime, one of the major obstacles is the lack of p-type $Ga_2O_3$ which can be utilized as the PN homo-junction termination for the BV improvement. It was calculated that shallow acceptor does not exist and it was also predicted that the holes are self-trapped inherently[6]. As a result, unipolar $Ga_2O_3$ power electronics dominate most of the research and few reports are available about the bipolar transport study. Due to the challenge of realizing p-type $Ga_2O_3$ on lightly-doped n-type $Ga_2O_3$ layer, the BV of the vertical $Ga_2O_3$ power diodes was limited, although various types of edge termination (ET) methods were employed[7]. On the other hand, some wide-bandgap p-type materials like NiO$_x$ with $E_g$ of 3.8–4 eV and Cu$_2$O with $E_g \sim 3$ eV, controllable doping and decent hole mobility of 0.5–5 cm$^2$/V s turn out to be a good counterpart of p-type $Ga_2O_3$ to boost the diodes performance[8,9]. The combination of p-NiO$_x$ and n-$Ga_2O_3$ is a feasible route for the $Ga_2O_3$ development, and the recent progress of the $Ga_2O_3$ PN hetero-junction (HJ) diodes shows a P-FOM of 1.37 GW/cm$^2$, which is comparable to the P-FOM value of state-of-the-art Schottky barrier diodes (SBDs)[10,11]. Even incorporating p-NiO$_x$ into $Ga_2O_3$ material system, the potential of $Ga_2O_3$ HJ PN diodes is only explored for less than 10% of the material limitation. Meanwhile, the conductivity modulation effect is observed in $Ga_2O_3$ HJ PN diodes, indicating the holes can

be injected in the $Ga_2O_3$ layer[12]. However, under what bias condition and to what extent the conductivity modulation can impact the $R_{on,sp}$ are still not explored. In addition, the in-depth understanding of bipolar transport in the $Ga_2O_3$ layer, especially hole transport and lifetime extraction are still forfeiting. Against the generally believed holes are self-trapped, the hole lifetime is a crucial and fundamental parameter to determine whether the bipolar transport is about to happen and to what extent it will impact the PN diode performances. Another critical issue regards the practical application of UWB PN diode is the requirement of low turn-on voltage ($V_{on}$) for high-efficiency application, since the general forward bias ($V_F$) is limited to be around 3 V. This is very challenging for homo-junction PN diode for wide bandgap semiconductor GaN and SiC with $V_{on} \sim 3$ V, regardless of the even wider bandgap $Ga_2O_3$.

In this article, a general design strategy of UWB semiconductor power diodes is provided to achieve high BV and low $R_{on,sp}$ simultaneously through the introduction of hole injection and transport in $Ga_2O_3$ to minimize the $R_{on,sp}$, suppressing the background carrier density to improve the BV, employing low conduction band offset p-NiO$_x$ to reduce $V_{on}$, and a composite E management technique with implanted ET and field plate architecture to further strengthen the BV. We setup a milestone of the UWB power diodes by acquiring a BV of 8.32 kV and P-FOM = BV$^2$/$R_{on,sp}$ of 13.21 GW/cm$^2$, which is a record P-FOM value among all types of UWB power diodes to date, and it also exceeds the 1-D unipolar limit of GaN and SiC. Meanwhile, a conductivity modulation phenomenon induced bipolar transport of electron and hole pairs is identified with hole lifetime determined to be 5.4–23.1 ns. Considering some real application circumstances of diodes at a $V_F$ of 3 V, benchmarking of the BV and $R_{on,sp}$ extracted at $V_F = 3$ V also shows a record P-FOM value to date, validating the great promise of UWB power diodes for next-generation high-voltage and high-power electronics.

## Results and discussions

**High BV and low $R_{on,sp}$ design strategy and implementation.** The most intriguing aspect of β-$Ga_2O_3$ is that its native substrate can be substantially grown by the melt-grown methodology, which lays a basic foundation for low-cost and large-diameter with low defect density substrate[13]. The β-$Ga_2O_3$ epi-layer can be epitaxied by various routes, such as molecular beam epitaxy, metalorganic chemical vapor deposition (MOCVD), mist-CVD, halide vapor phase epitaxy (HVPE), and some other low-cost techniques[14,15]. HVPE is the most widely adopted methodology for balancing the epitaxial speed, substrate size, defect density, and complicity. The β-$Ga_2O_3$ background doping regulation is a challenge, resulting in a non-controllable electron density of $2$–$4 \times 10^{16}$ cm$^{-3}$. Unintentional doping from precursors like Si or H, and defects like O vacancies all contribute to the n-type conduction in the β-$Ga_2O_3$ layer.

Ideal power devices should embrace high BV and low $R_{on,sp}$ to provide high blocking capability and low loss simultaneously[16]. In order to improve the BV of the UWB $Ga_2O_3$ power diodes, the minimal doping concentration is the first essential, since the slope of the E is governed by the doping concentration[16]. Summarized in Supplementary Fig. 1, it was found that the BV of the reported $Ga_2O_3$ power diodes is limited to be less than 3 kV, where the donor concentration is the major tyranny. Some other subsidiary factors like ETs or advanced E management techniques are all prerequisites for a minimized peak E at the anode edge to achieve a high desirable BV. PN junction is one of the most straightforward approaches to suppress the peak E at the interface. However, the forfeit of the p-$Ga_2O_3$ on the n-$Ga_2O_3$ makes the PN home-junction an impossible mission to further

explore the maximum BV potentials of diodes. It should be noted that only extending the spacing of two electrodes to increase the BV is of marginal value by sacrificing the $R_{on,sp}$ and averaged E. In terms of manipulating the $R_{on,sp}$, to increase the doping concentration seems to be the simplest, however, the BV will be essentially compromised. A unique physical phenomenon of power diode, which is called conductivity modulation of the PN or PIN junction at forward bias will substantially guarantee a low $R_{on,sp}$ even at a low doping concentration. Regardless of the challenge on the formation of PN homo-junction, the high $V_{on} > 4$ V is another suffering for the $Ga_2O_3$ homo-junction PN diodes.

Recently, the implementation of the p-$NiO_x$ into the $Ga_2O_3$ system opens up another route for expanding the $Ga_2O_3$ application from the SBDs to the HJ PN diode[17–20]. Although the performance of the $Ga_2O_3$ HJ diode is still inferior to the SBDs at current stage, however, we argue that some fundamental limitations which have haunted the $Ga_2O_3$ power diodes research for a decade could be essentially clarified. First, p-$NiO_x$ flavors a low conduction band offset of ~2.1 eV such that the high $V_{on}$ issue of the homo-junction could be partially resolved. Second, with a PN HJ structure, the conductivity modulation is theoretically expected so that the $R_{on,sp}$ can be minimized at a low doping concentration and high $V_F$. In addition, by combining the ETs and advanced E management, the BV can be further enhanced. Comparison of the E management strategies is summarized in Supplementary Fig. 2.

Figure 1a shows the 3-D cross-sectional image of two representative $Ga_2O_3$ HJ PN diodes, the top view image is exhibited as Fig. 1b, and the false-colored scanning electron microscopy (SEM) image at the crucial area of the anode edge is listed in Fig. 1c. In the $Ga_2O_3$ power diodes, the doping concentration of the $Ga_2O_3$ epi-layer is suppressed from the $2 \times 10^{16}$ cm$^{-3}$ to around $5–7 \times 10^{15}$ cm$^{-3}$ for two wafers with different thicknesses by adopting a long duration of the oxygen

thermal anneal process, as shown in Fig. 1d[21]. $C–V$ curves are shown in Supplementary Fig. 3. Heavily doped p-$NiO_x$ layer on top is utilized to form an Ohmic contact, as described in Fig. 1e. The simulated energy band diagram of the p-$NiO_x$/n-$Ga_2O_3$ HJ is shown in Fig. 1f with the conduction band and valance band offset to be 2.15 eV and 2.8 eV, respectively. The ET process by Mg doping to form a high-resistivity region underneath the electrode is utilized to withstand a high E and the coupled field plate is implemented to further mitigate crowded peak E at the anode edge[15].

**Diodes characterizations.** Figure 2a compares the log-scale forward current-forward bias-ideality factor ($I_F$-$V_F$-$\eta$) characteristics of two $Ga_2O_3$ HJ PN diodes with $Ga_2O_3$ thickness ($T_{Ga2O3}$) of 7.5 and 13 μm at a radius of 75 μm. The kink effect observed at $V_F$ around 1.5 V is related to the variation of the barrier height and ideality factor, which is most likely to be induced by the two different barriers connected in parallel. $I_F$ on/off ratio of $10^9$-$10^{10}$ and $\eta$ smaller than 2 can last for 4–5 decades of the $I_F$. Figure 2b shows the linear-scale forward $I_F$-$V_F$-$R_{on,sp}$ curves of the same diodes as Fig. 2a. Even with a PN HJ structure, a relatively decent $V_{on} = 1.8$ V is acquired, which is much smaller than the $V_{on}$ of SiC and GaN PN diodes. The small $V_{on}$ is benefited from two aspects, the small conduction band offset between p-$NiO_x$ and n-$Ga_2O_3$ and the interface recombination current[17]. Minimal Diff. $R_{on,sp}$ is extracted to be 2.9 and 5.24 mΩ cm$^2$ for $T_{Ga2O3} = 7.5$ and 13 μm, respectively. Unlike SBDs with increased $R_{on,sp}$ at an increased $V_F$, the $R_{on,sp}$ of the $Ga_2O_3$ HJ PN diodes drops at an increased $V_F$, most likely due to bipolar transport-induced conductivity modulation effect. It should be noted that such conductivity modulation effect is the key to enable the simultaneous achievement of low $R_{on,sp}$ and high BV. Figure 2c describes the radius-dependent $I_F$-$V_F$-$R_{on,sp}$ curves for diodes with $T_{Ga2O3} = 13$ μm. Log-scale $I_F$-$V_F$ characteristic is

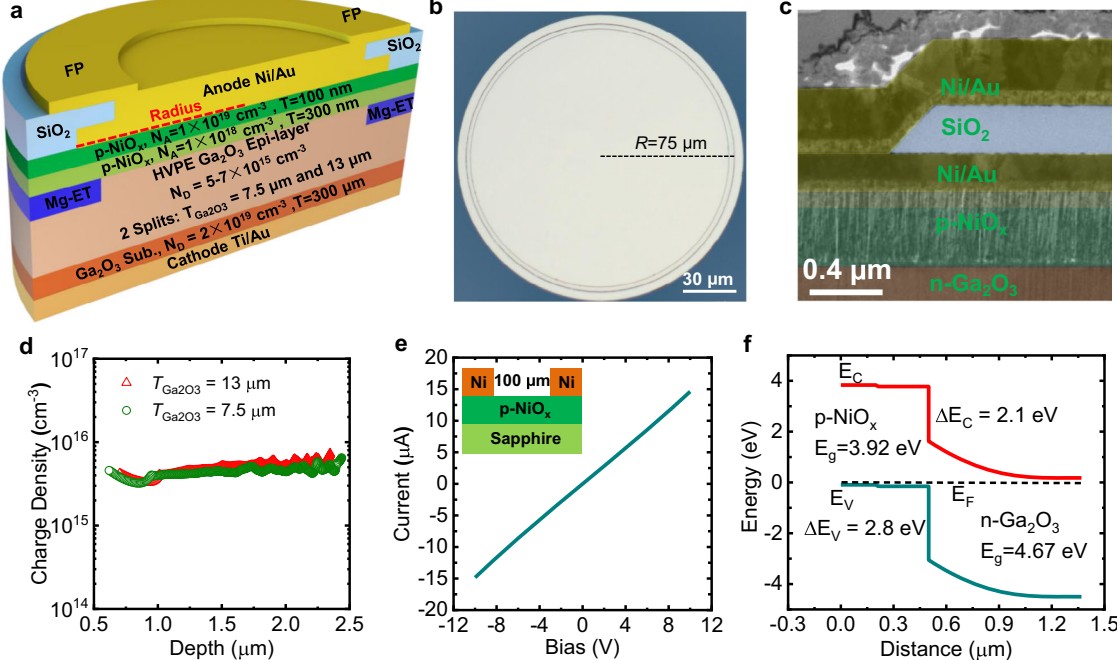

**Fig. 1 UWB power diodes design and implementation. a** 3-D cross-sectional schematic of the $Ga_2O_3$ power diodes with HJ architecture and composite electric field management. **b** Top view of a fabricated $Ga_2O_3$ power diode. **c** False-colored SEM image of the cross-sectional anode field plate region with p-$NiO_x$ thickness of 400 nm. **d** Extracted carrier concentration of two representative samples with concentration of $5 \times 10^{15}$–$7 \times 10^{15}$ cm$^{-3}$. **e** Current-voltage behavior of the Ni pads on p-$NiO_x$ with $N_A = 10^{19}$ cm$^{-3}$, showing an Ohmic contact. **f** Simulated band diagram of the p-$NiO_x$/n-$Ga_2O_3$ HJ structure. The band bending occurs in n-$Ga_2O_3$ and the conduction band offset is only 2.1 eV, showing the great promise of low $V_{on}$ even for a UWB material.

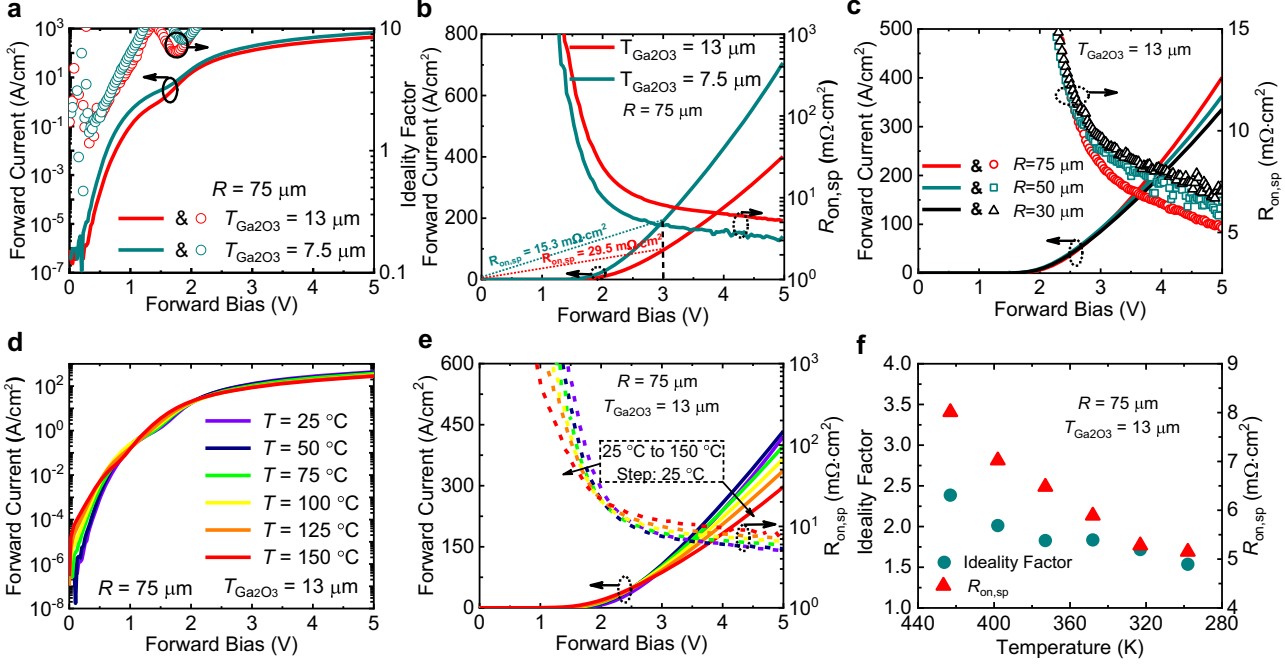

**Fig. 2 UWB Ga$_2$O$_3$ power diodes forward characteristics. a** Forward current-voltage-ideality factor characteristics of two Ga$_2$O$_3$ power diodes with $T_{Ga2O3} = 7.5$ and 13 μm. **b** Forward current–voltage-specific on-resistance $R_{on,sp}$ characteristics of the same diodes as **a**. A decent $V_{on} = 1.8$ V with minimal Diff. $R_{on,sp} = 2.9$ and 5.24 mΩ cm$^2$ as well as extracted overall $R_{on,sp}$ (@$V_F = 3$ V) of 15.3 and 29.5 mΩ cm$^2$ for $T_{Ga2O3} = 7.5$ and 13 μm are achieved. **c** Radius-dependent forward current-voltage-resistance curves for diodes with $T_{Ga2O3} = 13$ μm. T-dependent **d** log-scale and **e** linear-scale forward characteristics of diode with $T_{Ga2O3} = 13$ μm. On/Off ratio of 10$^{10}$ and 10$^8$ are achieved for $T = 25$ °C and 150 °C, respectively. At all temperatures, $R_{on,sp}$ drops when $V_F$ increases, verifying the conductivity modulation effect. **f** Extracted T-dependent ideality factor and $R_{on,sp}$ values as **e**.

summarized in Supplementary Fig. 4. By increasing the radius, the insulating Mg implanted region constitutes to a smaller portion of the area so that $R_{on,sp}$ decreases when radius increases. The resistance (Res.) contribution from each layer based on the equation Res. = thickness/($N_D \times \mu \times q$) is summarized in Supplementary Fig. 5[22]. For diodes with $T_{Ga2O3} = 7.5/13$ μm, $N_D = 6 \times 10^{15}$ cm$^{-3}$, $\mu = 200$ cm$^2$/Vs, and $q = 1.6 \times 10^{-19}$ C, the resistance of the drift layer is calculated to be 3.89/6.77 mΩ cm$^2$. It should be noted that this calculation is based on the low-level injection prerequisite. At $V_F = 5$ V, conductivity modulation effect of the HJ PN diode begins to dominate so that the $R_{on,sp}$ drops, which is favorable for resistance minimization. Figure 2d and e summarizes the T-dependent linear-scale $I$-$V$-$R_{on,sp}$ and log-scale $I_F$-$V_F$ of the diode with radius = 75 μm and $T_{Ga2O3} = 13$ μm. On/off ratios of 10$^{10}$ and 10$^8$ are achieved at $T = 25$ °C and 150 °C, respectively. At all temperature ranges, the differential (Diff.) $R_{on,sp}$ drops at an increased $V_F$, verifying the conductivity modulation effect of the Ga$_2$O$_3$ HJ PN diodes. Figure 2f shows the extracted T-dependent ideality factor $\eta$ and $R_{on,sp}$ from temperatures of 25–150 °C. The $\eta$ is extracted from the forward current equation $J = J_s(\exp(qV_F/\eta kT) - 1)$, whereas $J_s$ is the reverse saturation current, $V_F$ is the applied forward bias, $q$ is the electron charge, k is the Boltzmann's constant, and $T$ is the absolute temperature. The $\eta$ is extracted to be around 1.5 at $T = 25$ °C.

Based on the simulation, it is very interesting to find that hole concentration in the Ga$_2$O$_3$ layer at the HJ-interface is comparable with the Ga$_2$O$_3$ doping concentration of $6 \times 10^{15}$ cm$^{-3}$ at $V_F = 3.5$ V, as shown in Supplementary Figs. 6 and 7. That is to say, the hole injection-related conductivity modulation can help to reduce the $R_{on,sp}$ only with $V_F \geq 3.5$ V, since the hole is with more than 1 order of magnitude lower mobility. The simulation result is in good agreement with the forward $I_F$–$V_F$ characteristic, since the $R_{on,sp} = 7$ mΩ cm$^2$ (at

$V_F = 3.5$ V) roughly equals to the resistance summary of p-side Ohmic contact, p-NiO$_x$ layer, n-Ga$_2$O$_3$ drift layer, n$^+$-Ga$_2$O$_3$ substrate and n-side Ohmic contact. In other words, the hole injection and conductivity modulation are negligible at the $V_F$ range of $V_{on} = 1.8$ V to 3.5 V, due to significant valance band offset between p-NiO$_x$ and n-Ga$_2$O$_3$, so that few holes can be injected across this barrier. At $V_F \sim 3.5$ V, holes are injected from p-NiO$_x$ to n-Ga$_2$O$_3$ most likely via trap assisted tunneling and hopping mechanisms. By increasing the $V_F$ beyond 3.5 V to lower the PN HJ barrier, more holes are injected into n-Ga$_2$O$_3$ layer and hence high level injection phenomenon will raise the electron concentration in the Ga$_2$O$_3$ layer to maintain the charge neutrality condition. Therefore, the $R_{on,sp}$ is further reduced when the $V_F$ is increased. At $V_F = 5$ V, the hole concentration is simulated to be $3.8 \times 10^{16}$ cm$^{-3}$ and $6 \times 10^{15}$ cm$^{-3}$ at HJ-interface and 6 μm away from the HJ-interface, respectively. The averaged hole (also electron) concentration is extracted to be $1.9 \times 10^{16}$ cm$^{-3}$ within this 6-μm range, by integrating concentration and then divided by the total length of 6 μm. Therefore, the resistance of the significant hole injection region is roughly calculated to be 1.32 mΩ cm$^2$, by considering the electron mobility of 150 cm$^2$/Vs at this electron concentration. By adding up another 7-μm low level injected Ga$_2$O$_3$ layer resistance of 6.77/$13 \times 7 = 3.65$ mΩ cm$^2$, the 13-μm Ga$_2$O$_3$ drift layer owns a $R_{on,sp}$ of 4.97 mΩ cm$^2$. This estimation of the $R_{on,sp}$ coincides with our extracted $R_{on,sp}$ from the $I_F$–$V_F$, verifying the correctness of the explanation, hole concentration simulation, and calculation of the hole injection into the Ga$_2$O$_3$ layer.

The T-dependent reverse $I$–$V$ characteristics of diode with $T_{Ga2O3} = 7.5$ μm are plotted in Fig. 3a from $T = 25$–150 °C. By increasing the $T$, $I_R$ increases, indicating a non-avalanche breakdown behavior. Even at $T = 150$ °C, the $I_R$ is just 1 mA/cm$^2$ at a reverse bias of 3 kV. By further pushing the reverse bias to 5.1 kV we observe a hard breakdown with

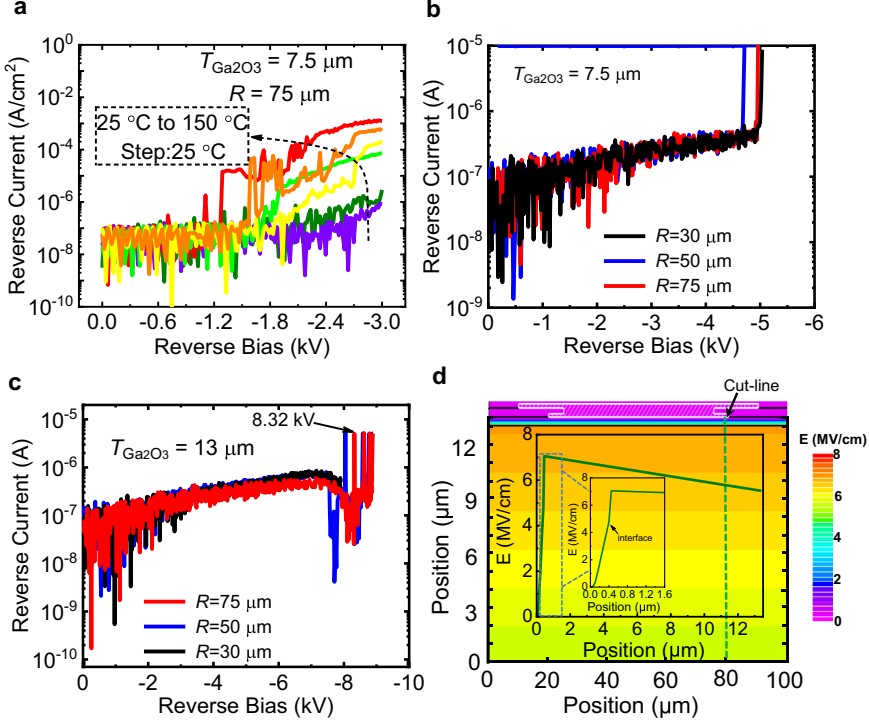

**Fig. 3 UWB Ga₂O₃ power diodes with high breakdown voltages. a** T-dependent reverse current–voltage characteristics of diode with $T_{Ga2O3} = 7.5\,\mu m$. With increased $T$, $I_R$ increases, indicating a non-avalanche breakdown. Room temperature reverse current–voltage characteristics of diodes with $T_{Ga2O3} = 7.5\,\mu m$ (**b**) and 13 μm (**c**) at various radiuses. A BV of 5.1 kV and 8.32 kV are achieved for diodes with $T_{Ga2O3} = 7.5$ and 13 μm, yielding an averaged $E$ of 6.45 MV/cm and 6.2 MV/cm, respectively. **d** Simulated E distribution of the diode with BV = 8.32 kV and $T_{Ga2O3} = 13\,\mu m$. Due to the small $N_D = 6 \times 10^{15}\,cm^{-3}$, a fully depletion and a small E slope of the drift layer is observed.

$T_{Ga2O3} = 7.5\,\mu m$, as indicated in Fig. 3b. The averaged $E$ field is calculated to be around 6.45 MV/cm by considering $E = 5.1\,kV/(0.4\,\mu m + 7.5\,\mu m)$. Combined with the $R_{on,sp} = 2.9\,m\Omega\,cm^2$, the P-FOM = $BV^2/R_{on,sp}$ is yielded to be 8.97 GW/cm². As for the diode with $T_{Ga2O3} = 13\,\mu m$, a maximum BV of 8.32 kV is acquired at an $I_R = 0.2\,mA/cm^2$, as exhibited in Fig. 3c. The as-measured figure is shown in Supplementary Fig. 8. This BV = 8.32 kV is the highest BV value among all Ga₂O₃ power FETs and diodes to date. As a result, the P-FOM is calculated to be $(8.32\,kV)^2/5.24\,m\Omega\,cm^2 = 13.21\,GW/cm^2$. Besides the record P-FOM, this HJ PN diode also has a high averaged $E = 8.32\,kV/(0.4\,\mu m + 13\,\mu m) = 6.2\,MV/cm$. Figure 3d describes the E simulation result of the HJ PND with $T_{Ga2O3} = 13\,\mu m$ and BV = 8.32 kV. The simulated peak E in the p-NiO$_x$ layer is around 4.9 MV/cm, which is slightly lower than its theoretical limit, considering the 3.9 eV bandgap. The peak E at the p-NiO$_x$ side is lower when compared with the peak $E$ at the Ga₂O₃ side, due to much higher dielectric constant of p-NiO$_x$. Due to the small $N_D$ and the depletion effect from the p-NiO$_x$ as well as the functionalities of the ET and coupled field plate, a fully depletion and small E slope are observed in the drift layer, resulting a peak $E = 7\,MV/cm$ in the Ga₂O₃ at the HJ-interface.

**Holes in Ga₂O₃ layer.** Similar to other UWB semiconductors like diamond, BN, and AlN, high ionization efficiency of n- and p-type doing simultaneously turns out to be a big challenge, considering the UWB nature of those UWB semiconductor materials. The direct observation of conductivity modulation is a straightforward evidence of bipolar transport and hole existence in the Ga₂O₃ layer, which deviates from the general prediction that holes are less likely to occur in Ga₂O₃. Three reasons are attributed to the challenge of acquiring holes in Ga₂O₃, no calculated shallow acceptors, large effective mass from the flat

valance band, and free holes tend to be self-trapped by polarons. However, we argue that with the unique PN HJ structure under high $V_F$ condition, holes from the heavily-doped p-NiO$_x$ are capable of being injected to the Ga₂O₃ layer, although the hole mobility is relatively low. Under a very positive $V_F$ condition (e.g., 5 V), energy band of the p-NiO$_x$ is pulled down so that holes at the Fermi tail witness no significant barrier height to travel across the PN HJ-interface and then diffuse in the Ga₂O₃ layer, leading to the conductivity modulation effect. In other words, the holes can be manufactured in the UWB Ga₂O₃ layer by hole injection at a very positive $V_F$. In order to verify the hole transportation and survival in the Ga₂O₃ not so short by self-trapping effect of the polarons, hole lifetime extraction or measurement is urgently needed.

The reverse recovery measurement technique is implemented to determine the hole lifetime in the Ga₂O₃ layer, and the schematic of the measurements are summarized in Supplementary Fig. 9a[23]. Once the Ga₂O₃ diode is switched from positive $V_F$ to a reverse bias, a period of time is needed to remove holes from the Ga₂O₃ either via electron-hole pair recombination or to be trapped by polarons. The hole lifetime ($\tau_p$) can be determined by the equation $\tau_p = t_{sd}/(erf^{-1}(I_F/(I_F + I_R)))^2$, whereas $t_{sd}$, $I_F$, and $I_R$ represent charge storage time, forward current, and reverse current, respectively[24]. During the reverse recovery measurement, the $V_F$ is extracted to be 2.97 V and 4.73 V for injection current of 5 mA and 25 mA, respectively, at a diode radius of 40 μm. Meanwhile, the subsequently applied reverse bias is −8 V. The reverse recovery and input–output measurement of the pulsed current–voltage characteristics of the UWB Ga₂O₃ HJ PN diode at a diode current of 5 mA is shown in Fig. 4a. For the HJ PN diode, the $I_F$ is 5 mA which is 3 orders of magnitudes more than $I_R$, so that $I_F/(I_F + I_R)$ can be simplified to be 1. Then the $\tau_p$ can be simplified to $\pi/4 \times t_{sd}$, which is ~80% of the $t_{sd}$ when the diode

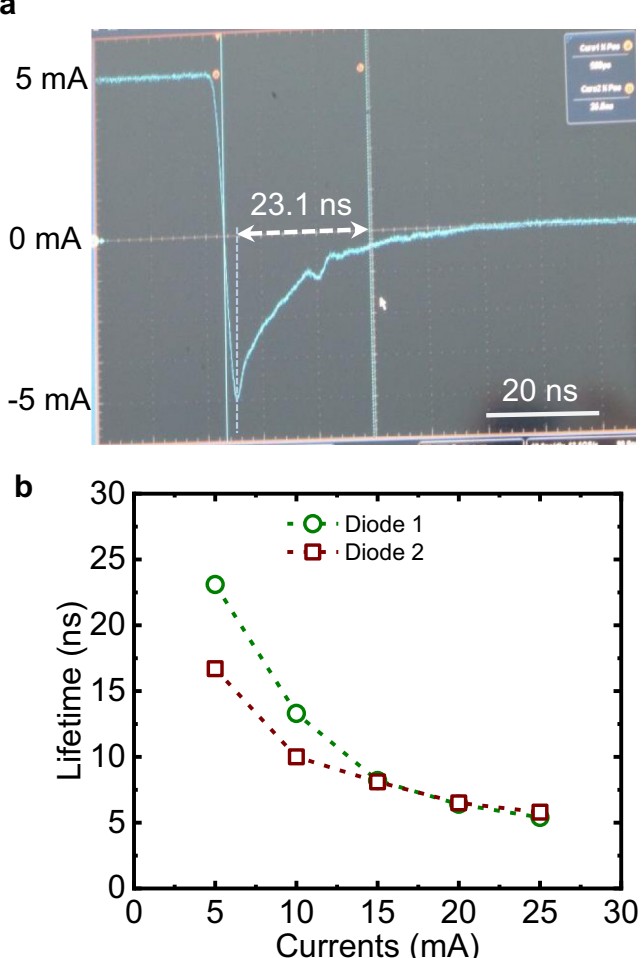

**Fig. 4 Hole lifetime determination in Ga₂O₃ layer. a** Time-dependent of the reverse recovery characteristics of Ga₂O₃ HJ PN diode at a forward injection current of 5 mA. **b** Lifetime dependence on the forward injection current with current ranges from 5 mA to 25 mA. At current of 5 mA, the lifetime is determined to be 23.1 ns. At high $V_F$ condition, holes diffuse from p-NiO$_x$ to n-Ga₂O₃ without seeing obvious barrier, so that the hole lifetime in the Ga₂O₃ layer can be determined.

is switched until the anode current is recovered to be around 0. Therefore, hole lifetime $\tau_p$ is determined to be 23.1 ns at a forward injection current $I_F$ of 5 mA. The $\tau_p$ dependence on $I_F$ is summarized in Fig. 4b, with a minimal $\tau_p$ of 5.4 ns. In order to exclude the subsidiary impact on the measurements, the reverse recovery measurement is performed on Ga₂O₃ SBD (Supplementary Fig. 9b) and the recovery time in the SBD is determined to be 1.8 ns, which is 1 order of magnitude lower when compared with the HJ PN diode. By injecting holes into the Ga₂O₃ layer at high $V_F$, the hole lifetime is then determined to be 5.4–23.1 ns. By combining the calculated hole effective mass ($m_p^*$) of $4.46m_o$ (Supplementary Fig. 10), the hole mobility ($\mu_p$) can be roughly estimated by the equation $\mu_p = q \times \tau_p/m_p^*$, yielding the $\mu_p$ to be 1.93–8.3 cm²/Vs.

**Performance benchmarking**. The combination of the conductivity modulation induced low $R_{on,sp}$ and low doping concentration as well as the composite E regulation led record BV renders a substantial performance enhancement by setting a record P-FOM of all UWB power diodes (Fig. 5a), including Ga₂O₃, diamond, and high Al-Al$_x$Ga$_{1-x}$N ($x > 60\%$) power diodes[25–44]. Compared with all other Ga₂O₃ power diodes, the BV of this work is around three times the

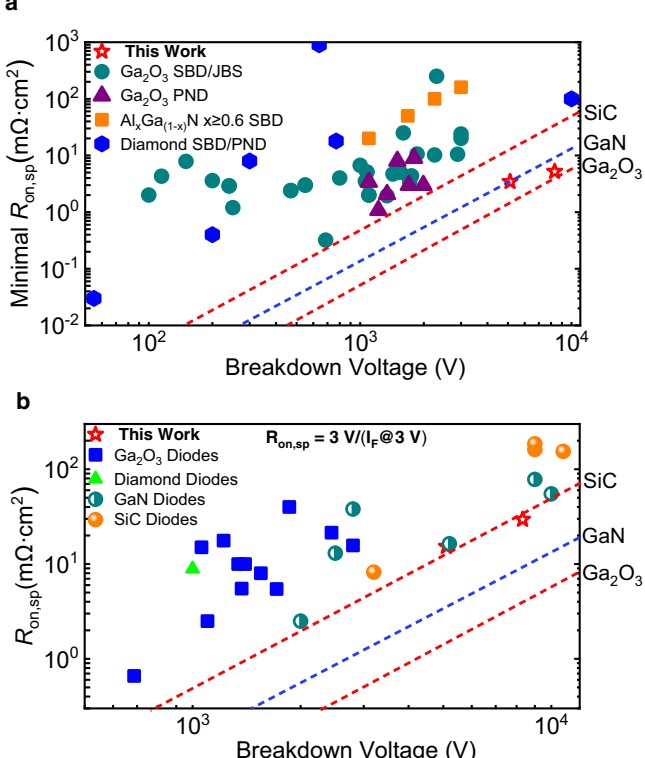

**Fig. 5 Benchmarking UWB Ga₂O₃ power diodes against state-of-the-art other diodes. a** Minimal $R_{on,sp}$ versus BV of some representative UWB power diodes, including Ga₂O₃, diamond, and high-Al AlGaN, which are reported in the literatures. Our Ga₂O₃ power diodes set a milestone for the UWB power diodes by breaking the 1-D unipolar figure-of-merit limit of GaN and SiC. **b** Extracted $R_{on,sp}$(@3V) versus BV of some highest performance GaN, SiC, and diamond diodes. By considering some real application circumstances, the $R_{on,sp} = 3V/(I_F@3 V)$ is preferred over the minimal $R_{on,sp}$ to eliminate the impact of the $V_{on}$. GaN and SiC PN diodes are excluded due to the $V_{on} \sim 3$ V. Our UWB power diodes demonstrate a substantial enhancement of the performance over other diodes by surpassing the 1-D unipolar limit of the SiC. The $R_{on,sp}$ extraction for lateral diodes is yielded by $R_{on,sp} = $ on-resistance × (anode–cathode spacing +1.5 μm transfer length for both electrodes).

previously reported best BV of 2.9 kV with a lower $R_{on,sp}$. The most enticing aspect of this work is that the performance of the Ga₂O₃ device exceeds the 1-D unipolar limit of the SiC and GaN. In terms of real application of the HJ PN diode in the circuit, the overall $R_{on,sp}$ at a general $V_F = 3$ V instead of the minimal differential $R_{on,sp}$ is more realistic. In order to eliminate the impact of $V_{on}$, an overall $R_{on,sp}$ of 15.3 mΩ cm² and 29.5 mΩ cm² are extracted for $T_{Ga2O3}$ of 7.5 μm and 13 μm at a $V_F = 3$ V, respectively, as shown in Fig. 2b. Benchmarking against all other state-of-the-art representative diodes, including SiC SBDs/JBS diodes/PN diodes and GaN SBDs/PN diodes with extracted $R_{on,sp}$ at a fixed $V_F = 3$ V, our Ga₂O₃ HJ PN diodes achieve a record of nowadays power diodes, as compared in Fig. 5b[33,45–53]. Even under the real application circumstance, the P-FOM = BV²/$R_{on,sp}$ of the Ga₂O₃ power diodes still surpasses the 1-D unipolar limit of the SiC. These intriguing results verify the great promise of UWB semiconductor Ga₂O₃ power diodes for next-generation high-voltage and high-power electronics.

In summary, we show that UWB semiconductor Ga₂O₃ power diodes are capable of delivering a record high BV²/$R_{on,sp}$, which breaks the 1-D unipolar limit of the SiC and GaN figure-of-merit. The incorporation of suppressed background doping, HJ PN structure, and the composite electric field management technique

yields a high BV which makes the averaged electric field approach the material limit. Taking advantage of the hole injection as well as the conductivity modulation, the $R_{on,sp}$ can be essentially minimized even the $Ga_2O_3$ is with a low doping concentration. The hole lifetime is determined to be 5.4–23.1 ns, which verifies the existence of the hole in the $Ga_2O_3$ layer. By carefully engineering the energy band offset, a decent $V_{on}$ can also be derived for a high efficiency rectifying. This unique technology by implementing the low doping material, electric field suppression, hole injection as well as the conductivity modulation, and energy band engineering offers an effective route for the innovation of other UWB power diodes, such as diamond, BN, high Al mole fraction $Al_xGa_{1-x}N$.

## Methods

**Fabrication of UWB $Ga_2O_3$ power diodes.** $Ga_2O_3$ epi-wafers with epi-layer thicknesses of 7.5 μm and 13 μm were epitaxial by HVPE on a (001) substrate with substrate doping concentration of $2 \times 10^{19}$ cm$^{-3}$. Substrates were first thinned down from 650 μm to 300 μm by polishing to minimize on-resistance. Then, $Ga_2O_3$ epi-wafers were annealed in the low-pressure-CVD furnace at 500 °C under the $O_2$ ambient to partially compensate the donors in the epi-layer. N-side Ohmic contacts were formed by evaporating Ti/Au metals followed with rapid thermal anneal at 450 °C. Angle-dependent Mg ion implantation was utilized to form a high-resistivity layer to serve as the ET. Bi-layers of p-$NiO_x$ were sputtered at room temperature with first and second layer doping concentration of $1 \times 10^{18}$ cm$^{-3}$ and $1 \times 10^{19}$ cm$^{-3}$, respectively. The doping concentration of the p-$NiO_x$ layer was confirmed by the Hall measurements and the Hall mobility of the second p-$NiO_x$ layer is 1.1 cm$^2$/Vs. P-side Ohmic contacts were formed by depositing Ni/Au layers. The field plate was constructed by depositing 300 nm of $SiO_2$, $SiO_2$ etching, and field plate metal evaporation. A summary of the device process schematic flow is shown in Supplementary Fig. 11.

**Device characterizations.** The forward I–V and C–V characteristics were carried out by the Keithley 4200 semiconductor analyzer systems. Reverse I–V measurements were performed by Agilent B-1505A high voltage semiconductor analyzer systems with extended high-voltage module up to 10 kV. The hole lifetime measurements were carried out by reverse recovery measurement methods as Supplementary Fig. 9.

## Data availability

The data that support the plots within this paper and other findings of this study are available from the corresponding author upon reasonable request. The reason for controlled access is due to privacy issue. The data is available from the corresponding author H.Z. for research purposes and the corresponding author will send the data within one week once received the request.

## Code availability

The simulation code that supports the plots within this paper and other findings of this study are available from the corresponding author upon reasonable request. The reason for controlled access is due to privacy issue. The data are available from the corresponding author H.Z. for research purposes and the corresponding author will send the data within one week once received the request.

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

## Acknowledgements

The work was supported by National Natural Science Foundation of China under the grant nos. 62004147 and 61925404. We are grateful to Dr. Y. Huang from Xidian Wuhu research institute for the 10-kV high voltage breakdown measurements. We are also grateful to Prof. Y. H. Zhang for the valuable discussions.

## Author contributions

H.Z. and J.C.Z. conceived the idea and proposed the $Ga_2O_3$ HJ PN diodes. Y.H. supervised the whole process and designed the experiment. P.F.D. did the device fabrication and some I-V characterizations. K.D., Y.N.Z., and Q.L.Y. carried out the breakdown measurements, taking SEM and TEM pictures, wafer packaging. H.X., Z.H.L., and M.W.S. carried out the hole lifetime measurements and set-up, partial I–V measurements, and partial data analyze. J.S. performed the DFT calculations. J.C.G. and M.F.K. performed the hole injection TCAD simulation. H.Z. and J.C.Z. co-wrote the manuscript and all authors commented on it.

## Competing interests

The authors declare no competing interests.
