## [Peer Review File · Nature Communications]

Title: Ultra-Wide Bandgap Semiconductor Ga₂O₃ Power DiodesREVIEWER COMMENTS

Reviewer #3 (Remarks to the Author):

Review:

This work proposes an improvised hetero junction p-NiOx /Ga2O3 diode with state of art lower Von, higher BV (8.32KV) and lower Ron,sp by implementing hole injection in the Ga2O3 layer causing bipolar transport/conductivity modulation. The results overall are impressive. The validation of the physical mechanisms (bipolar transport/conductivity modulation) claimed could be detailed better

Minor corrections:

1. Page 1-6, The comparison of the SiC, GaN, Ga2O3 metrics like BFOM, PFOM, mobility, critical E field, bang gap etc, could be better represented in a table in introduction in addition to the prose. Cu2O, p-NiOx metrics could also be added (if needed).

2. Page 6 “ a composite E management technique with implanted ET” E ? ET ? expand before using shorthand notations

3. Page 8 “P-NiOx flavours” ◇ “P-NiOx favours”
“ET process by Mg doping” ◇ add a reference

Clarifying questions:

a) Page 10

Ideality factor can be a good indication for surfaces in homojunctions. Will the same diode model be a good indicator for heterojunctions? Also mention which diode equation is used to extract the factor. At higher temperatures till 380K ideality factor is relatively constant and increases beyond that what is the inference with respect to the surfaces from this observation ?

b) The HJ PN structure demonstrated, have a lower Ron,sp at higher bias attributed to the conductivity modulation

- Can you validate this? Showing carrier concentrations in Ga2O3 layer at such bias conditions in TCAD simulation can be helpful

- The measured higher lifetimes based on the reverse recovery measurements can also be a characteristic of a high quality Ga2O3 (low recombination) and changes in injection condition. How would you decouple this attributing it to bipolar conduction only

Reviewer #4 (Remarks to the Author): 
see file attached

Reply to the Reviewer's Comments

The authors appreciate the insightful comments from all the Reviewers. We addressed all comments from the Reviewers and Editor and made the revision. The following is the response to the Reviewers' comments point by point.

Reviewer Comments:

Reviewer 3

This work proposes an improvised hetero junction p-NiO_x/Ga₂O₃ diode with state of art lower V_{on}, higher BV (8.32KV) and lower R_{on,sp} by implementing hole injection in the Ga₂O₃ layer causing bipolar transport/conductivity modulation. The results overall are impressive. The validation of the physical mechanisms (bipolar transport/conductivity modulation) claimed could be detailed better.

Thanks for the general positive comments. We have resolved all the concerns related with the physical mechanism.

Minor corrections:

1. Page 1-6, The comparison of the SiC, GaN, Ga₂O₃ metrics like BFOM, PFOM, mobility, critical E field, bang gap etc, could be better represented in a table in introduction in addition to the prose. Cu₂O, p-NiO_x metrics could also be added (if needed).

Thanks for this comment. The comparison between SiC, GaN, and Ga₂O₃ metrics are summarized in Table 1 of the supplementary information (SI). In addition, Cu₂O and p-NiO_x are also compared in the Table 2 of the SI. The details are listed as below.

Table 1: Comparisons of SiC, GaN and Ga₂O₃ power semiconductor materials

Materials	4H-SiC	GaN	β-Ga ₂ O ₃
Bandgap E _g (eV)	3.25	3.4	4.8
Dielectric Constant ε	10	9	9-10
Breakdown Field E _C (MV/cm)	2.5	3.3	8
Electron Mobility μ (cm ² /Vs)	1000	1200	300
Saturation Velocity v _{sat} (10 ⁷ cm/s)	2	2.5	2
Thermal Conductivity κ (W/mK)	370	250	10-30
FOM relative to Si			
Baliga FOM = ε×μ×E _c ³	317	846	3200
Johnson FOM = E _c ² ×v _{sat} ² /4π ²	278	1089	2844
Baliga High Frequency FOM = μ×E _c ²	46	100	142
Keyes FOM = κ×[(c×v _{sat})×(4π×ε)] ^{1/2}	3.6	1.8	0.2

Table 2: Comparisons of p-NiO_x and p-Cu₂O

Materials	NiO _x	Cu ₂ O
Bandgap E _g (eV)	3.7-4	~3
Dielectric Constant ε	11.9	8.27
Breakdown Field E _C (MV/cm)	4.3-5	~2.8
Hole Mobility μ (cm ² /Vs)	5	2.7

2. Page 6 “ a composite E management technique with implanted ET” E ? ET ? expand before using shorthand notations

Thanks for this comment. E represents “electric field” and ET represents “edge termination”.

In the revised draft, we have expanded the notation at their first occurrence in the manuscript, e.g. at page 3 and page 4.

3. Page 8 “P-NiO_x flavours” “P-NiO_x favours”

“ET process by Mg doping” add a reference

Thank you for this comment. The reference is added as Ref. [15] in the revised manuscript at page 9.

Clarifying questions:

a) Page 10

Ideality factor can be a good indication for surfaces in homojunctions. Will the same diode model be a good indicator for heterojunctions? Also mention which diode equation is used to extract the factor. At higher temperatures till 380K ideality factor is relatively constant and increases beyond that what is the inference with respect to the surfaces from this observation ?

Thanks for this comment. Yes, the reviewer is absolutely correct that the same model is also a good indicator for heterojunction. The underline physics is that the forward bias is used to control the barrier height and the forward current is then determined by the barrier height, regardless of whether it is a homojunction or heterojunction. The equation $J=J_s(\exp(qV/\eta kT)-1)$ is used to extract the ideality factor, where J_s is the reverse saturation current, V is the applied forward bias, q is the electron charge, η is the ideality factor, k is the Boltzmann’s constant, and T is the absolute temperature. This equation is universal for different types of diodes (Schottky, homo or heterojunction PN diode) at forward bias condition, whereas the barrier height change is correlated with the forward bias V by the ideality factor η . Therefore, the same model can be applied to the heterojunction case. The ideality

factor η increases with the increase of the temperature, which is very similar to the subthreshold swing (SS). As described in Fig. 2(f), the overall trend of η is that the η increases with the T. As for the reviewer's comment "At higher temperatures till 380K ideality factor is relatively constant and increases beyond that what is the inference with respect to the surfaces from this observation", we believe it is caused by this particular diode and the η increase magnitude before 380 K is not as significant as the η increase magnitude after 380 K. From a detailed glance, η still increases when the T is increased from 298 K to 380 K. Meanwhile, we have provided a statistic study about the η from 5 diodes, the results are shown as below. The averaged η increases along with the increase of the T.

Figure 1: Extracted ideality factor η versus temperature from 5 diodes. The general trend of η is that the η increases with the increase of the T.

In the revised draft, we have added "*The ideality factor η is extracted from the forward current equation $J=J_s(\exp(qV_F/\eta kT)-1)$, whereas J_s is the reverse saturation current, V_F is the applied forward bias, q is the electron charge, k is the Boltzmann's constant, and T is the absolute temperature.*" at page 10.

b) The HJ PN structure demonstrated, have a lower $R_{on,sp}$ at higher bias attributed to the conductivity modulation

- Can you validate this? Showing carrier concentrations in Ga_2O_3 layer at such bias conditions in TCAD simulation can be helpful.

Thanks for this comment. Based on the TCAD simulation, the hole concentration versus the depth at various forward bias conditions are shown as below. During the simulation, heterojunction trap assisted tunneling and hopping, band to band tunneling, thermionic emission, Auger recombination, SRH recombination, and direct tunneling models are all utilized. It is noticed that at forward bias (V_F) of 3 V, the hole injection is not significant. As the V_F increases to 3.5 V, the hole injection becomes obvious. The $R_{on,sp}$ (@ $V_F=3.5$ V) extracted from I_F-V_F is slightly smaller than the calculated resistance ($7.3 \text{ m}\Omega \cdot \text{cm}^2$) from the summary of the p-side Ohmic contact, p-type NiO layer, $n^-Ga_2O_3$ layer, $n^+Ga_2O_3$ substrate and n-side Ohmic contact, indicating the occurrence of the obvious hole injection and conductivity modulation. At $V_F = 4-5$ V,

the hole concentration is simulated to be 10^{16} - $4 \times 10^{16} \text{ cm}^{-3}$ at the heterojunction interface, which is significantly higher than the electron doping concentration in Ga_2O_3 layer. Therefore, the $R_{\text{on,sp}}$ drops as the V_F increases. It should be noticed that only when $V_F \geq 3.5 \text{ V}$, then the obvious hole injection and conductivity modulation are expected to happen.

Figure 2: Simulated Ga_2O_3 hole concentration versus depth from the HJ-interface at various forward bias conditions.

In the revised draft, the hole simulation figure is added as the Figure 7 of SI.

In the revised draft, we have also added, “Based on the simulation, it is very interesting to find that hole concentration in the Ga_2O_3 layer at the HJ-interface is comparable with the Ga_2O_3 doping concentration of $6 \times 10^{15} \text{ cm}^{-3}$ at $V_F = 3.5 \text{ V}$, as shown in Supplementary Figure 6 and 7. That is to say, the hole injection related conductivity modulation can help to reduce the $R_{\text{on,sp}}$ only with $V_F \geq 3.5 \text{ V}$, since the hole is with more than 1 order of magnitude lower mobility. The simulation result is in good agreement with the forward I_F - V_F characteristic, since the $R_{\text{on,sp}} = 7 \text{ m}\Omega \cdot \text{cm}^2$ (at $V_F = 3.5 \text{ V}$) roughly equals to the resistance summary of p-side Ohmic contact, p- NiO_x layer, n⁻- Ga_2O_3 drift layer, n⁺- Ga_2O_3 substrate and n-side Ohmic contact. In other words, the hole injection and conductivity modulation are negligible at the V_F range of $V_{\text{on}} = 1.8 \text{ V}$ and 3.5 V , due to significant valance band offset between p- NiO_x and n- Ga_2O_3 , so that few holes can be injected across this barrier. By increasing the V_F beyond 3.5 V to lower the PN-HJ barrier, more holes are injected into n- Ga_2O_3 layer and hence high level injection phenomenon will raise the electron concentration in the Ga_2O_3 layer to maintain charge neutrality condition. Therefore, the $R_{\text{on,sp}}$ is further reduced when the V_F is increased.”

- The measured higher lifetimes based on the reverse recovery measurements can also be a characteristic of a high quality Ga_2O_3 (low recombination) and changes in injection condition. How would you decouple this attributing it to bipolar conduction only.

Thanks for this comment. In order to clarify this possibility, two steps are utilized. First of all, we have fabricated the Schottky barrier diode (SBD) with the same epi-wafer as the HJ-PN diode. High temperature annealing, implanted edge

termination, and field plate structure are all included. SBD also has a breakdown voltage around 5 kV, verifying the high quality of the Ga₂O₃ material. The reverse recovery measurement was performed on the SBD, and the effective carrier lifetime is determined to be 1.8 ns, as can be found in Figure 9 of the Supplementary Information. When the anode is switched from forward bias condition to the reverse bias condition, electrons in the Ga₂O₃ layer are likely to be swept out from the Ga₂O₃ layer via the cathode electrode, since the cathode electrode is with higher potential. As for HJ-PN diode case, when the anode is switched from forward bias to reverse bias, holes will move toward anode because of the low potential. However, holes will see a significant barrier at the HJ-interface and the only way for holes to disappear is by recombining with the electrons or being captured by traps. This is how reverse recovery measurement works. Therefore, the effective lifetime extracted from reverse recovery measurement is about the hole lifetime. Secondly, the general trend for reverse recovery measurement is that carrier has a higher lifetime as the injection and carrier density is less, as verified and shown in many PIN diodes, as shown in Fig. 2, Ref. [1] and [2]. Therefore, at smaller injection current or smaller carrier concentration conditions, the extracted higher effective carrier lifetime belongs to the representative characteristics of the PIN diodes.

Figure 3: Extracted carrier lifetime dependence on the injection current from Reference 2. In general, the lower injection current (lower concentration) the higher extracted lifetime.

References:

- [1] Zheng, D. W., Smith, B. T., Dong, J., Asghari, M., On the effective carrier lifetime of a silicon p-i-n diode optical modulator. *Semicond. Sci. Technol.* 23, 064006 (2008).
- [2] Caverly, R. H. et al. SPICE Modeling of Microwave and RF Control Diodes. In *Proc. 43rd IEEE Midwest Symp. on Circuits and System*, Lansing MI, Aug 8-11 (2000).

We would like to appreciate the reviewer again to help us understand the HJ-diode comprehensively and improve the quality significantly.

Reviewer 4

Key results:

In the manuscript, “Ultra-Wide Bandgap Semiconductor Ga₂O₃ Power Diodes” the authors report for the first time a power figure of merit that exceeds published values of GaN or SiC. This is achieved by a Ga₂O₃ drift layer with very low doping density and conductivity modulation due to hole injection via a NiO/ Ga₂O₃ pn-heterojunction.

Thanks for the general positive comments and acknowledging the state-of-the-art high performance diodes.

Data and methodology:

- Band diagram calculated for forward bias conditions to validate that hole injection from NiO to Ga₂O₃ is possible.

Thanks for this comment. We have simulated the band diagram at forward bias (V_F) of 5 V, as shown in Fig. 1(a), where the effective barrier for holes to inject from p-NiO_x to n-Ga₂O₃ layer becomes negligible compared to that at low V_F conditions. Therefore, holes are expected to be injected from p-NiO_x side to n-Ga₂O₃ since there is almost no barrier for holes. Meanwhile, the simulated hole concentration in the Ga₂O₃ layer at various V_F conditions are shown as Fig. 1(b). In the simulation, at $V_F = 3V$, the hole injection is not significant so that the conductivity modulation is not significant. Therefore, the $R_{on,sp}$ @ $V_F = 3V$ is larger than the calculated total resistance of p-side Ohmic contact, p-NiO_x layer, n-Ga₂O₃ layer, n⁺-Ga₂O₃ substrate and n-side Ohmic contact. At $V_F = 3.5 V$, hole concentration at the HJ-interface is comparable with the electron density so that the conductivity modulation begin to reduce the $R_{on,sp}$. By further increasing V_F beyond 3.5 V to 4 V or even 5 V, high level injection will raise electron density and significant conductivity modulation induced $R_{on,sp}$ drop phenomenon can be observed.

Figure 1: Simulated (a) energy band diagram of p-NiO_x and n-Ga₂O₃ HJ at $V_F = 5 V$, (b) Ga₂O₃ hole concentration at various V_F conditions.

- The kink in the forward current (around 1.5V) should be explained, does this

implicate the existence of two regions with distinct barrier height?

Thanks for this comment. The kink effect can also be reflected as the change of the ideality factor η . In general, for a PN diode the closer the η is to 1, the higher the diffusion current component, and the closer to 2, the greater the proportion of the recombination current. The kink or the two stage phenomenon in the forward current at forward bias of 1.5 V is most likely related with the trap assisted carrier recombination, which is usually observed in different types of diodes. Those trap states can either locates at the p-NiO_x/n-Ga₂O₃ interface, or the Mg-implanted n-Ga₂O₃ interface.

In the revised draft, we have added, “The kink effect observed at V_F around 1.5 V is most likely related with the trap assisted carrier recombination, which is likely to be induced either by the p-NiO/n-Ga₂O₃ interface or originate from the Mg-doped n-Ga₂O₃ layer.” at page 9.

- Could the effect of the two different barrier height regions be caused by the Mg doped layer used for minimizing the E-field at the contact edge?

Thanks for this comment. Yes, the reviewer is correct that Mg doped layer might be one of the reasons since implantation will cause some defects so that the trap assisted recombination will change the ideality factor and the kink effect is then observed. Further optimization includes thermal annealing to heal those implanted defects.

In the revised draft, we have added “Further thermal annealing will essentially minimize those defects and traps, and hence minimizing the kink effect.” at page 9.

- Depict log-scale forward current for different radii

Thanks for this comment. The log-scale forward current for different radii is shown as below.

Figure 2: Radius dependent log-scale forward I-V characteristics.

In the revised draft, the radius dependent forward I-V characteristic is added as the Figure 4 of the Supplementary Information.

- Relate V_{on} to the band diagram

Thanks for this comment. The relatively small V_{on} is primarily determined by the small conduction band offset and recombination current at forward V_F , as explained by reference [1].

In the revised draft, the small conduction band offset is marked in the band diagram. In addition, we have also added, “The small V_{on} is benefited from two aspects, the small conduction band offset between p-NiO_x and n-Ga₂O₃ and the interface recombination current [17].” at page 9.

- Flat conduction bands are expected for 2.1V, close to the measured turn-on voltage, the valence band offset is still significant at V_{on} , however, since the diode is already operated under flat band conditions, authors should explain, by what mechanism holes are injected into Ga₂O₃. In reference 9 (given below) it was concluded that hole injection into Ga₂O₃ is not significant for the large valence band offset between NiO and Ga₂O₃ – please comment.

Thanks for this comment. As indicated in reference 9, the paper wrote, “Our previous study revealed a large valence band offset of ~ 2.3 eV at the heterojunction [14], which is a high barrier to holes and could effectively block them from flowing into the β -Ga₂O₃ drift region even when the diode is turned on.”. Of course, the reviewer is correct that at $V_F = V_{on} \sim 2$ V and when the diode is turned on, the V_F is not enough to cover the valence band offset, so that the hole injection is less likely to occur. However, we propose that by continuing increasing V_F beyond certain value, the valence band can be pulled down and those holes can be injected to Ga₂O₃ layer.

We have simulated energy band diagram at $V_F = 3.5$ V and 5 V, as shown in Fig. 3(a) and 3(b). Meanwhile, the simulated Ga₂O₃ hole concentration versus depth in the Ga₂O₃ layer is shown as Fig. 3(c). The $V_F = 3.5$ V is chosen because we find that $V_F = 3.5$ V is around the turning point from low level injection to the high level injection. At $V_F = 3.5$ V, there is a 0.65 eV E_V difference between p-NiO_x and n-Ga₂O₃. Then holes can travel across this barrier by trap assisted tunneling and hopping via trap states, which are close to the E_V . We have performed deep level transient spectrum (DLTS) characterizations and the trap states are found to be around 10^{16} - 10^{17} cm⁻³ close to the E_V . Therefore, at the medium $V_F \sim 3.5$ V range, the hole injection mechanism is likely to be related with trap assisted tunneling and hopping from the p-NiO_x to n-Ga₂O₃. By continuing increasing the V_F , the E_V of the p-NiO_x is further pulled down so that there is no obvious barrier for holes to diffuse across the HJ-interface. Therefore, under high V_F circumstance (e.g. ≥ 4 V), hole injection mechanism is related with hole diffusion without significant barrier.

In order to verify the possible hole injection mechanism analysis, we have carried out TCAD simulation by combining the heterojunction trap assisted tunneling and hopping, band to band tunneling, thermionic emission, Auger recombination, SRH

recombination, and direct tunneling models. In addition, a theoretical calculation based on I_F - V_F characteristic is also implemented to roughly match the simulation result. Based on our calculation, without considering the hole injection, the 13- μm drift layer has a resistance of $6.7 \text{ m}\Omega\cdot\text{cm}^2$ and the total diode resistance is around $7.3 \text{ m}\Omega\cdot\text{cm}^2$. Only when the V_F is greater than 3.5 V , the $R_{\text{on,sp}}$ becomes smaller than the $7.3 \text{ m}\Omega\cdot\text{cm}^2$, and the hole injection and conductivity modulation is suggested to reduce total resistance. Actually, from the simulation result, we also noticed that the hole concentration at the HJ-interface is comparable with the electron density in the Ga_2O_3 layer at $V_F = 3.5 \text{ V}$. In addition, at $V_F = 5 \text{ V}$, the hole concentration is extracted to be $3.8 \times 10^{16} \text{ cm}^{-3}$ and $6 \times 10^{15} \text{ cm}^{-3}$ at HJ-interface and $6 \mu\text{m}$ away from the HJ-interface, respectively. The averaged hole (also electron) concentration is extracted to be $1.9 \times 10^{16} \text{ cm}^{-3}$ within this $6\text{-}\mu\text{m}$ range, by integrating concentration and then divided by the total length of $6 \mu\text{m}$. Therefore, the resistance of the significant hole injection region is roughly calculated to be $1.32 \text{ m}\Omega\cdot\text{cm}^2$, by considering the electron mobility of $150 \text{ cm}^2/\text{Vs}$ at this electron concentration. By adding up another $7\text{-}\mu\text{m}$ low level injected Ga_2O_3 layer resistance of $6.77/13 \times 7 = 3.65 \text{ m}\Omega\cdot\text{cm}^2$, the $13\text{-}\mu\text{m}$ Ga_2O_3 drift layer owns a $R_{\text{on,sp}}$ of $4.97 \text{ m}\Omega\cdot\text{cm}^2$. This estimation of the $R_{\text{on,sp}}$ coincides with our extracted $R_{\text{on,sp}}$ from the I_F - V_F , verifying the correctness of the explanation, hole concentration simulation and hole injection mechanism.

In summary, holes are injected from p- NiO_x to n- Ga_2O_3 through two kinds of mechanisms. At medium $V_F \sim 3.5 \text{ V}$, holes are injected via trap assisted tunneling and hopping mechanisms. At high $V_F \sim 5 \text{ V}$, holes diffuse from p- NiO_x to n- Ga_2O_3 since there is no significant barrier.

Figure 3: Simulated energy band diagram at $V_F =$ (a) 3.5 V and (b) 5 V , (c) simulated Ga_2O_3 hole concentration versus distance from HJ-interface at various V_F conditions.

We have added the energy band diagram and hole concentration simulation figures as the Figure 6 and Figure 7 of the Supplementary Information.

In the revised draft, we have added a whole paragraph to explain the hole injection and conductivity modulation mechanisms, “Based on the simulation, it is very interesting to find that hole concentration in the Ga_2O_3 layer at the HJ-interface is comparable with the Ga_2O_3 doping concentration of $6 \times 10^{15} \text{ cm}^{-3}$ at $V_F = 3.5 \text{ V}$, as shown in Supplementary Figure 6 and 7. That is to say, the hole injection related

conductivity modulation can help to reduce the $R_{on,sp}$ only with $V_F \geq 3.5$ V, since the hole is with more than 1 order of magnitude lower mobility. The simulation result is in good agreement with the forward I_F - V_F characteristic, since the $R_{on,sp}=7$ m Ω ·cm² (at $V_F = 3.5$ V) roughly equals to the resistance summary of p-side Ohmic contact, p-NiO_x layer, n-Ga₂O₃ drift layer, n⁺-Ga₂O₃ substrate and n-side Ohmic contact. In other words, the hole injection and conductivity modulation are negligible at the V_F range of $V_{on} = 1.8$ V and 3.5 V, due to significant valance band offset between p-NiO_x and n-Ga₂O₃, so that few holes can be injected across this barrier. At $V_F \sim 3.5$ V, holes are injected from p-NiO_x to n-Ga₂O₃ most likely via trap assisted tunneling and hopping. By increasing the V_F beyond 3.5 V to lower the PN-HJ barrier, more holes are injected into n-Ga₂O₃ layer and hence high level injection phenomenon will raise the electron concentration in the Ga₂O₃ layer to maintain charge neutrality condition. Therefore, the $R_{on,sp}$ is further reduced when the V_F is increased. At $V_F = 5$ V, the hole concentration is extracted to be 3.8×10^{16} cm⁻³ and 6×10^{15} cm⁻³ at HJ-interface and 6 μ m away from the HJ-interface, respectively. The averaged hole (also electron) concentration is extracted to be 1.9×10^{16} cm⁻³ within this 6- μ m range, by integrating concentration and then divided by the total length of 6 μ m. Therefore, the resistance of the significant hole injection region is roughly calculated to be 1.32 m Ω ·cm², by considering the electron mobility of 150 cm²/Vs at this electron concentration. By adding up another 7- μ m low level injected Ga₂O₃ layer with resistance of $6.77/13 \times 7 = 3.65$ m Ω ·cm², the 13- μ m Ga₂O₃ drift layer owns a $R_{on,sp}$ of 4.97 m Ω ·cm². This estimation of the $R_{on,sp}$ coincides with our extracted $R_{on,sp}$ from the I_F - V_F , verifying the correctness of the explanation, hole concentration simulation and calculation of the hole injection into the Ga₂O₃ layer.”.

- What is the influence of high interface recombination currents as reported in ref. 8 (given below)

Thanks for this comment. The high interface recombination current is useful for the diode turn on. As shown in the band diagram of Fig. 1f, the conduction band offset is 2.1 V, and there is some depletion in the Ga₂O₃ layer, so that the turn on voltage should be much larger than 1.8 V. However, due to the interface recombination current, the V_{on} of the diode is reduced from 3 V to 1.8 V. The ideality factor of 2 at $V_F = 1.5$ V is a good indicator of the interface recombination current.

In the revised draft, we have added “The small V_{on} is benefited from two aspects, the small conduction band offset between p-NiO_x and n-Ga₂O₃ and the interface recombination current [17].” at page 9.

- Compare on-resistance to recent literature values

Thanks for this comment. The $R_{on,sp}$ -BV comparison with state-of-the-art diodes is summarized in Fig. 5a and Fig. 5b. For power switching application, small on-resistance is preferable; however, high BV is also an essential. In order to get a low on-resistance, high doping concentration material could be used. However, this

high doping approach will sacrifice the BV. Based on our understanding, both low on-resistance and high BV are required for a high performance power diode. Although our $R_{on,sp}$ is similar or slightly higher than other Ga_2O_3 HJ-PN diodes, however, our BV is around 4-8 times of other Ga_2O_3 HJ-PN diodes. Therefore, comparing and combining on-resistance and BV in the same figure seems to be more meaningful.

- Figure 3 d should magnify the interface, being the most interesting part of the structure.

Thanks for this comment. We have magnified the interface point and the revised figure is shown as below.

Figure 4: Simulated E distribution of the diode with $BV = 8.32$ kV and $T_{Ga_2O_3} = 13$ μm .

- The initial current of the reverse recovery characteristics is with 5mA rather low.

Please provide V_F and the subsequently applied reverse bias used during reverse recovery measurement

Thanks for this comment. The reverse recovery measurements are performed on a much smaller diode (radius = 40 μm), so that the current looks like a little bit low. The reverse recovery measurements are carried out by injecting a current, which is a general route to do this kind of measurement. The forward bias V_F at 5 mA injection current is measured to be 2.97 V, and the V_F is measured to be 4.73 V at a 25 mA injection current. The applied reverse bias of -8 V is used during the recovery measurement.

In the revised draft, we have added “During the reverse recovery measurement, the V_F is extracted to be 2.97 V and 4.73 V for injection current of 5 mA and 25 mA, respectively, at a diode radius of 40 μm . Meanwhile, the subsequently applied reverse bias is -8 V.” at page 14.

Approach and logic order of the manuscript:

The manuscript lacks a discussion of latest results on p-NiO/n-Ga₂O₃ power heterojunction diodes as well as heterojunction barrier Schottky diodes, e.g., in the introduction and in the experimental results for comparison of own finding to the current state-of-the-art. Further, first reports on hole injection and conductivity modulations are not discussed nor cited. A list of recent publications is included below. Further, the rather general introduction to the material Ga₂O₃ could be shortened, to provide space for discussion of recent Ga₂O₃ power diodes.

Thanks for this comment. We have double checked with those 9 references, references 2, 3, 5, 8 have been cited in our manuscript as the representative progress about HJ-PN diode, references 6 and 7 were published after the submission of our manuscript, and references 1, 4, 9 are similar same-group works as what we have cited in our manuscript. As suggested by the reviewer, we have cited all of them in our manuscript and those newly added references are marked as red.

As for the conductivity modulation in those papers, all of them just mention this effect, however, none of them provide an in-depth analysis, calculation or simulation of related holes. Therefore, in our submitted manuscript (first version), we have this sentence “Even incorporating p-NiO_x into Ga₂O₃ material system, the potential of Ga₂O₃ HJ-PN diodes is only explored for less than 10% of the material limitation. The in-depth understanding of bipolar transport in the Ga₂O₃ layer especially hole transport and lifetime extraction are still forfeiting.”

We have deleted partial of the Ga₂O₃ introduction at page 6. The added information regards the most recent progress about heterojunction PN diode and conductivity modulation are listed as below, “Meanwhile, the conductivity modulation effect is observed in Ga₂O₃ HJ-PN diodes, indicating the holes can be injected in the Ga₂O₃ layer [12]. However, under what bias condition and to what extent the conductivity modulation can impact the R_{on,sp} are still not discussed.” at page 5.

In addition, we have added a whole paragraph to explain the hole injection and conductivity modulation.

Originality and significance:

Conductivity modulation (see Ref. 6 below) and very low specific on-resistance in p-NiO/n-Ga₂O₃ heterojunction power diodes literature (see Ref. 5-9 below) was already. Further, the advanced switching properties of NiO/Ga₂O₃ have been characterized and reported (ref. 7-9). The significance of the current manuscript is the demonstration of the first Ga₂O₃-based power diode with higher figure of merit P-FOM than GaN or SiC based diodes.

Thanks for this comment. The reviewer is absolutely correct that the one significant aspect of this paper is the demonstration of high performance Ga₂O₃ diode with record P-FOM value by acquiring high BV and low R_{on,sp} simultaneously, which is more than several times of other Ga₂O₃ HJ-PN diodes. **As for the originality and significance, we want to emphasize that we have proposed a general route for ultra-wide bandgap semiconductor power diodes to improve their performances.**

First of all, we observed the conductivity modulation effect and provided hole injection mechanisms to realize bipolar transport, which are never described in other Ga₂O₃ papers. Then, with the conductivity modulation, it is possible that we can minimize the carrier concentration without compromising the R_{on,sp}. By suppressing the background carrier concentration and combining the advanced ETs, a BV = 8.3 kV is acquired. As far as all the authors knowledge, other Ga₂O₃ HJ-diode papers only mention there is a phenomenon of conductivity modulation, however, they didn't take full advantage of this phenomenon to further improve their HJ-diodes performance. They didn't try to minimize the carrier concentration and then utilize conductivity modulation to get a low R_{on,sp}, while maintaining a high BV at the same time. Finally, as mentioned in our draft, this design strategy can be implemented to other ultra-wide bandgap power diodes, since achieving both the n and p type doping is a challenge. Therefore, beside from the record performance, our draft provides a general route for ultra-wide bandgap power diodes to increase the BV while maintaining a small R_{on,sp} simultaneously.

In the revised draft, we have strengthened the originality and significance of this work by adding, *“In this article, a general design strategy of UWB semiconductor power diodes is provided to achieve high BV and low R_{on,sp} simultaneously through the introduction of hole injection and transport in Ga₂O₃ to minimize the R_{on,sp}, suppressing the background carrier density to improve the BV, employing low conduction band offset p-NiO_x to reduce V_{on}, and a composite E management technique with implanted ET and field plate architecture to further strengthen the BV.”* at page 5.

Language:

The language of the manuscript should be revised. There are several awkward constructions that affect readability and in parts comprehensibility of the manuscript. I strongly suggest proof reading by a native speaker.

The authors should introduce any abbreviation and symbol used in the manuscript.

Thanks for this comment. We have sent our draft to a native speaker to revise the draft. We have defined those symbols at their first occurrence in the manuscript, e.g. page 3 and page 4.

Conclusions:

The reported breakdown voltage, specific on-resistance and power figure of merit are remarkable and define the state-of-the-art of Ga₂O₃-based power diodes. This result itself is not sufficient to justify publication in Nature Communication. Further, the manuscript does not contain sufficient citations of recently published relevant literature. Hence, results are not discussed and interpreted with adequate reference. The mechanism of hole injection should be discussed with respect to literature in which hole injection from NiO to Ga₂O₃ is found to be negligible. The language of the manuscript requires a detailed revision. I suggest rejection of the manuscript in its present state.

Thanks for this comment.

Here, the authors would like to more clarify the significance of this work, we believe our work is novel and provides an important new route toward high performance ultra-wide bandgap PN power diode demonstration as discussed below.

The authors agree with the reviewer that p-NiO/n-Ga₂O₃ heterojunction power diodes have been reported and conductivity modulation with low specific on-resistance have been observed. However, both on-resistance and breakdown voltages are required to achieve high P-FOM. In previous literatures, the high doping concentration in n-Ga₂O₃ contributes to a low on-resistance, which also causes reduced breakdown voltage, so that the impact of conductivity modulation on P-FOM is not significant. And how to balance breakdown voltage and on-resistance for high P-FOM is not discussed in previous works. In this work, we have provided a key route to enhance the performance p-NiO/n-Ga₂O₃ HJ power diode, that is, **to reduce the doping concentration of n-Ga₂O₃ for high breakdown voltage while not sacrificing on-resistance by conductivity modulation mechanism**. This was not reported or understood before. And we believe this is an important understanding toward high P-FOM, with which we have demonstrated record high P-FOM value among all ultra-wide bandgap power diodes.

In summary, in the revised manuscript, 1) we have demonstrated record ultra-wide bandgap power diode with highest P-FOM, 2) we have strengthened the originality and significance of this work by providing a general route for ultra-wide bandgap semiconductor power diodes to achieve high BV and low $R_{on,sp}$ simultaneously, which is not reported in other ultra-wide bandgap papers, 3) we have provided possible hole injection mechanisms to explain the V_F dependent conductivity modulation, which is forfeiting in other Ga₂O₃ HJ-diode papers, 4) draft is carefully revised and more references are included based on the reviewer guidance. We hope that this version can satisfy the high standard and requirement of the Nature Communication.

References:

[1] Lu, X. et al. 1-kV sputtered p-NiO/n-Ga₂O₃ heterojunction diodes with an ultra-low leakage current below 1 μ A/cm². IEEE Electron Device Lett., **41**, 449–452 (2020).

We would like to appreciate the reviewer again to help us understand the HJ-diode comprehensively and improve the quality significantly.

REVIEWERS' COMMENTS

Reviewer #3 (Remarks to the Author):

The questions are addressed by the authors

Reviewer #4 (Remarks to the Author):

The authors addressed the questions and comments and provided additional experimental and modelled results that improved the quality of the manuscript. There remain few points to be considered in a further revised version.

1. New TCAD simulation:

For the simulation the authors considered various transport mechanisms including heterojunction trap assisted tunneling and hopping, band to band tunneling, thermionic emission, Auger recombination, SRH recombination, and direct tunneling. Please discuss which of these mechanisms is mainly contributing to the hole injection observed for $V > 3.5V$. If trap assisted tunneling and hopping is the leading mechanism, please comment on the assumptions made (input parameters chosen) for modeling.

2. Kink in the forward bias direction

The creation of trap states within the Mg-implanted region must not be the origin of kink. The current at which the kink occurs depends systematically on the contact area. This systematic dependence of the current density observed at the kink with respect to the total current density for large forward bias indicates that a systematic change of the area ratio of the Mg-implanted and the active inner circle of the diode is the potential origin. This would be in agreement with the parallel diode model of Defives et al., in case the area ratio of the implanted and non-implanted parts of the diode indeed change for the diodes with different total contact area. Please check and comment.

Paper on Parallel Diode model

D. Defives, O. Noblanc, C. Dua, C. Brylinski, M. Barthula, V. Aubry- 368
Fortuna, and F. Meyer, "Barrier inhomogeneities and electrical character- 369
istics of Ti/4H-SiC Schottky rectifiers," IEEE Trans. Electron Devices, 370
vol. 46, no. 3, pp. 449–455, Mar. 1999.

3. Comment on the DLTS measurement

The net doping density of p-NiO and n-Ga₂O₃ are different by orders of magnitude (one-sided junction). Hence, the depletion layer can be assumed to be only within the Ga₂O₃ and a DLTS measurement would detect only defects within Ga₂O₃. It is therefore not clear to which valence band the authors refer by stating "...trap states are found to be around 10^{16} - 10^{17} cm⁻³ close to the EV." Please discuss if the trap states observed are minority of majority carrier traps. Is the result different, if one of the Schottky barrier diodes is investigated?

The manuscript can be accepted for publication after minor revision.

Reply to the Reviewer's Comments

The authors appreciate the insightful comments from all the Reviewers. We addressed all comments from the Reviewers and Editor and made the minor revision. The following is the response to the Reviewers' comments point by point.

Reviewer Comments:

Reviewer 3

The questions are addressed by the authors.

Thanks for accepting our manuscript.

Reviewer 4

The authors addressed the questions and comments and provided additional experimental and modelled results that improved the quality of the manuscript. There remain few point to be considered in a further revised version.

Thanks for the comment. We have replied those points based on the reviewer's suggestion and guidance.

1. New TCAD simulation:

For the simulation the authors considered various transport mechanisms including heterojunction trap assisted tunneling and hopping, band to band tunneling, thermionic emission, Auger recombination, SRH recombination, and direct tunneling. Please discuss which of these mechanisms is mainly contributing to the hole injection observed for $V > 3.5V$. If trap assisted tunneling and hopping is the leading mechanism, please comment on the assumptions made (input parameters chosen) for modeling.

Thanks for this comment. Yes, the reviewer is correct that the leading mechanism is the trap assisted tunneling and hopping. Trap levels with 0.1-0.7 eV above valence band of the Ga_2O_3 at a density of 10^{16} - 10^{17} cm^{-3} are assumed in the simulation. When the forward bias (V_F) is around 3.5 V-4 V, the valence band holes on the p-NiO side will tunnel to the trap levels of n- Ga_2O_3 , and then hopping to the valence band of n- Ga_2O_3 to form hole injection. It should be noted that when the V_F is greater than 4 V or 5 V, valence band of the p-NiO is lower than the valence band of n- Ga_2O_3 , so that holes can directly diffuse from p-NiO to the n- Ga_2O_3 and no hoping is needed.

In the revised draft, we have added "*At $V_F \sim 3.5$ V, holes are injected from p-NiO_x to n-Ga₂O₃ most likely via trap assisted tunneling and hopping mechanisms.*"

We have also updated the assumptions in the Supplementary Information.

2. Kink in the forward bias direction

The creation of trap states within the Mg-implanted region must not be the origin of

kink. The current at which the kink occurs depends systematically on the contact area. This systematic dependence of the current density observed at the kink with respect to the total current density for large forward bias indicates that a systematic change of the area ratio of the Mg-implanted and the active inner circle of the diode is the potential origin. This would be in agreement with the parallel diode model of Defives et al., in case the area ratio of the implanted and non-implanted parts of the diode indeed change for the diodes with different total contact area. Please check and comment.

Paper on Parallel Diode model D. Defives, O. Noblanc, C. Dua, C. Brylinski, M. Barthula, V. Aubry-Fortuna, and F. Meyer, "Barrier inhomogeneities and electrical characteristics of Ti/4H-SiC Schottky rectifiers," IEEE Trans. Electron Devices, vol. 46, no. 3, pp. 449–455, Mar. 1999.

Thanks for this comment. After double checking this reference paper mentioned by the reviewer and comparing our results, we agree with the reviewer that the kink effect is most likely related with parallel diode model with two barrier heights for the following two reasons. First of all, the barrier height and ideality factor are different, indicating the existence of two different barriers. Second, by increasing the temperature, a transition from the obvious "kink" effect to the "normal" behavior is observed, as evidenced in Fig. 2d of the manuscript. Those two factors of our diode completely coincide with the phenomenon mentioned in the reference.

In the revised draft, we have "~~deleted~~" the explanation of the "kink" effect, which is attributed to the implantation induced defect. Instead, we have added, "*The kink effect observed at V_F around 1.5 V is related with the variation of the barrier height and ideality factor, which is most likely to be induced by the two different barriers connected in parallel.*"

3. Comment on the DLTS measurement

The net doping density of p-NiO and n-Ga₂O₃ are different by orders of magnitude (one-sided junction). Hence, the depletion layer can be assumed to be only within the Ga₂O₃ and a DLTS measurement would detect only defects within Ga₂O₃. It is therefore not clear to which valence band the authors refer by stating "...trap states are found to be around 10^{16} - 10^{17} cm⁻³ close to the E_V ." Please discuss if the trap states observed are minority of majority carrier traps. Is the result different, if one of the Schottky barrier diodes is investigated?

Thanks for this comment. The trap states are located close to the valance band of the Ga₂O₃, as mentioned by the reviewer that the major depletion happens at the Ga₂O₃ side. Therefore, the observed states are minority carrier traps for HJ PN diode. As for Ga₂O₃ Schottky barrier diodes (SBDs), electron injection is expected to be observed so that only majority traps can be observed for SBDs. Similar minority and majority carrier trap types are observed in recent publication: Ye et al., IEEE Trans. on Electron Devices, vol. 69, no. 3 pp. 981-987, 2022.

In the revised supplementary information, we have mentioned the carrier trap states

to be "*Ga₂O₃ minority carrier traps*" at the simulation part figure 7.

The manuscript can be accepted for publication after minor revision.

Thanks for this comment. We have addressed all the concerns through this minor revision process.